# Combining Flood Risk Mitigation and Carbon Sequestration to Optimize Sustainable Land Management Schemes: Experiences from the Middle-Section of Hungary's Tisza River

### Gábor Ungvári

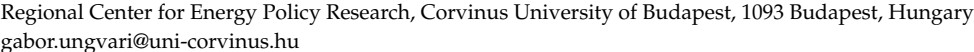

Regional Center for Energy Policy Research, Corvinus University of Budapest, 1093 Budapest, Hungary; gabor.ungvari@uni-corvinus.hu

**Abstract:** The record floods experienced along the Tisza River between 1998 and 2001 brought a paradigm shift in infrastructural solutions for flood protection. A flood peak polder system was built for transient water storage without any substantial change in land use in the polders, despite the potential to do so under the new scheme. The recent improvement of quantified flood risk assessment methodologies and stronger foundations for the valuation of carbon sequestration benefits now provide more information on the magnitude of missed opportunities and the potential for comprehensive land use and flood risk management solutions. This paper evaluates and combines the results of three cost-benefit type analyses on the conflicting relations of pursuing flood risk mitigation and land management goals. Although the studies were conducted at different locations of the same river stretch, they are all inspected using the same flood waves. Results assert that as EU-CAP agricultural subsidies stabilize individual benefits from arable land use in the short-run, public benefits and long-term individual benefits fail to reach their potential value. The combined analysis of flood risk change and $CO_2$ sequestration provides the economic rationale for the ecological revitalization along rivers with flood peak polders, helping to solve the conflict between hydrological and ecological objectives in floodplains. Capitalizing the value of the community benefits of forests in terms of $CO_2$ sequestration is limited by the unresolved property rights allocation of this natural capacity between landowners and the state, the latter being responsible for fulfilling international $CO_2$ reduction agreements; this uncertain legal background is an obstacle to the creation of sustainable economic conditions for the development and expansion of beneficial land management processes along rivers.

**Keywords:** flood risk management; $CO_2$ sequestration; cost-benefit analysis; agriculture; forestry

## 1. Introduction

The steady increase in flood risk is a widespread phenomenon [1]. Flood risk has two components, both of which strengthen this trend. On the social side, the economic value exposed to floods is increasing [2]. Regarding the probability that a flood occurs, the effect of climate change and deteriorating catchment conditions are reflected. As in many other regions, a shift in the pattern of rainfall events is observed in Central Europe [3]. Even without a change in annual precipitation, we can expect more concentrated precipitation events with occasionally higher discharge volumes [4]; these drivers force the implementation of new flood protection solutions. The "spatial flood risk management" approach partly responds to these challenges [5]; it focuses on the reconciliation of the natural and socio-economic conditions of areas that have the capacity to mitigate floods. The need to integrate additional land into floodplains arises because, typically, the land along rivers that is still available for floods, even supplemented with the defense capacities (levies), can no longer provide adequate peak-discharge capacity. At the same time, the approach aims at creating higher quality environmental conditions along rivers, valued

more and more by society [6]; this is reflected in the legislative expectation for the joint implementation of the EU Floods Directive and the Water Framework Directive; however, integrated implementation typically lacks robustly applicable public policy solutions. This analysis examines the potential and limitations of such integration, using the example of the middle section of the Tisza River, from the perspective of how specific elements of economic assessment methodology can be applied to establish land-use change processes that are considered justified from a policy perspective, while rarely implemented.

At the turn of the millennium, the countries of Central Europe faced an unprecedented series of high-magnitude floods [7], which led to the reconsideration of defense strategies; this process unfolded along the Hungarian stretch of the Tisza River as well, setting a new course to flood risk management. The preferred method of raising dikes has been replaced by a multi-pronged strategy led by flood-peak polders that manage floods with a return frequency of 100 years or more, with dikes to be developed to the previous design standards, and the restoration and maintenance of run-off capacities in the floodplain (2004/LXVII Act on the Further Development of the Vásárhelyi Plan, hereinafter referred to as VTT); this change, on the one hand, can be seen as a paradigm shift due to the advances in flood management, but on the other hand, there has been no substantial shift in the use of the floodplain or the areas within the polders [8], which was an explicit objective of the VTT development plans and a driving force behind similar European processes, for example, nature-based solutions [6].

Opinions differ widely on the use of areas protected by flood defense infrastructure from rivers, including where water management and flood damage infrastructure is most effective in developing and preserving public and private interests [9,10]; this issue is of particular importance in the lowland section of the Tisza, where de-flooding has confined the river to a particularly narrow area compared to other European rivers [11]. It is now clear that the large-scale socio-economic development initiated in the first half of the 19th century transforming the floodplain generated long-term costs being felt today [12].

The expansion of supply type agricultural production has degraded regulation-type ecosystem services that could handle, for example, the increased frequency of water extremes [13]. The question is how the economic monetization of these regulating services [14] can support the reorganization of lands to improve social welfare. The areas connected to the current floodplain either by dike relocations or flood-peak polders already contribute to reducing flood risk, irrespective of their current land-use, which can be considered a service to the people living in the former floodplain of the river [15,16]; however, the potential for introducing additional regulating-type ecosystem services is already land-use dependent. Increasing the total individual and community benefits with knowledge of the underlying natural science is fundamentally a public policy challenge [5].

This paper aims to reveal the economic specifics of this policy challenge and proceedings that have the potential to overcome this challenge.

## 2. Data and Methods

Over the past decade and a half, a polder system was developed with new large flood-peak polders along the Tisza River and the smaller shallow flood-emergency sites along the tributaries were upgraded with floodgates. (see Figure 1). Research programs have quantified the flood risk reduction impact of these interventions on the Middle and Upper Tisza.

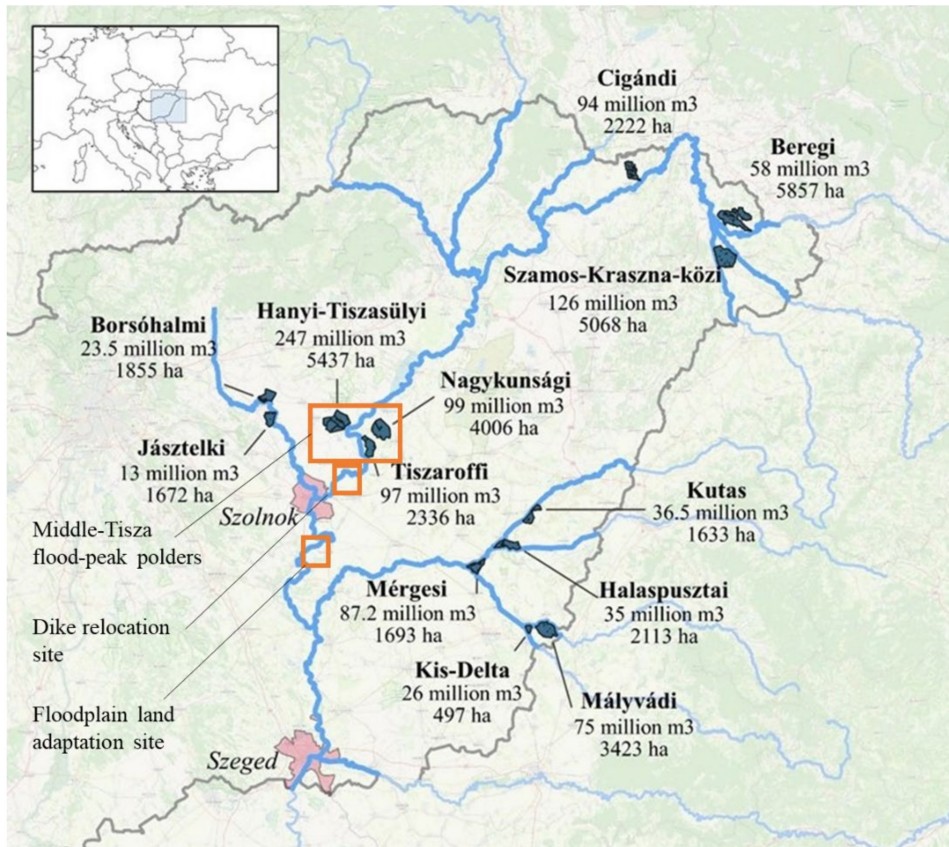

**Figure 1.** The middle section of the Tisza River. Legend: Map shows the flood polder system of the area and highlights the location of the study areas. Adapted from Ref. [17].

The economic analyses presented in this article are made possible by the improved methods of quantified flood risk assessment [18,19] and the quantification of the benefits of ecosystem services, in particular the mitigation of climate change through $CO_2$ sequestration [7]. Besides infrastructure development, these are the aspects that have the highest potential to shape the decisions on the ground. The interrelations are analyzed with the cost-benefit approach, considering whether transformed land uses in river corridors and former floodplain areas have the economic backing to cover the costs of change towards sustainable multipurpose land management forms.

This article draws on and develops the results of three analyses [20–22]; all three are an integral part of previous works led by the author. The added value of this article is that it presents, based on the results of these three sub-analyses carried out for different purposes, a coherent methodological approach that allows a quantifiable expression of the public interest and a financial comparison with the land use value, optimized for pursuing private interests. The approach builds on the most accepted economic value elements of land use and flood risk management. The application of the approach demonstrates that, even without the elements of ecological services that are difficult to quantify, there is sufficient information to support a positive economic equilibrium for land use solutions that provide higher public benefits and are financially sustainable, as a prerequisite for the transformation of private uses, while integrating profit-seeking activities.

The first report [20] explores the economic value of the flood peak reduction effect of the controlled opening of polders and an article based on it [17] presents the methodology of the calculation; it introduces a breakeven point flood frequency approach, above which (for less frequent, more severe flood events) the opening of a polder can be considered economically justified under current rules. In contrast to this, the present article examines the relationship between more frequent (that means lower peaking) floods and the use

of the polders below the breakeven point; this is the probability segment of floods where the conflicts between the public and private use of floodplains (due to the presence or absence of frequent flooding) are concentrated. The second report used [21] calculates the economic balance of a dike relocation intervention and gives detailed account of the steps of the calculation process; it provides the basis for comparing the magnitude of the benefits, the different ecosystem services that can be realized in case of the polders' controlled and the dike relocation's uncontrolled inundation. It reveals the different ecosystem services' contribution to the economic balance.

The third analysis [22] aimed to explore the economic outcomes of different land use and flood regimes on a former floodplain area. It uses the same methodology as seen in Ref. [21]. The land-use scenarios identified in the analysis did not produce results sufficiently characteristic to further consider their economic aspects. Therefore, this paper builds on the results of an analysis of an additionally prepared scenario assuming full afforestation, in order to interpret the scale of the benefits from $CO_2$ sequestration.

## 3. Results

### 3.1. Description of the Status Quo

This chapter describes the transformation of flood defense strategy along the Tisza River that created the infrastructure and the potential that points towards a sustainable land use on the floodplains. The status quo description explains the nature of the policy barriers. The results subsequently presented will outline the economic context identified for realizing this potential.

In the early 2000s when the review of the flood defense strategy began, experts did not have the technology and information background to quantify the impact of development alternatives for flood risk [23]. By the mid-2010s, when the flood peak polder system was already in place, these technical conditions had been satisfied under the EU Floods Directive; it is this guidance and progress that allows for our cost-benefit economic analysis.

Flood-peak polders became necessary when it was discovered that the cross-sectional runoff capacity of the river corridor (the area between the two dikes) cannot be adapted to the long-run trend of rising flood peaks using the current dikes [24]. The development of increasingly higher peaks from even similar discharge volumes is influenced by the deteriorating runoff conditions in the catchments and by the floodplain filling with sediment load from the watershed; this latter process has caused a significant rise in the ground level of the floodplain (1.5–2 m) since the construction of the dikes in the mid-19th century [25]. Consequently, the long-term loss of flood defense capacity is an inherent feature. In the short-term (a few decades or even years), the runoff cross-section can be further degraded by increasingly thick vegetation, which slows down the flow and the resulting backwater that causes flood levels to rise [26,27]. The resulting increase in flood risk will exacerbate conflicts of interest about land management decisions in the floodplain between flood protection, nature conservation, agriculture and forestry. All these stakeholders have only partially compatible demands for land management and maintenance. The interventions to curb natural processes increase the risk of the spread of invasive species [28], which in turn have a feedback effect, worsening flood risk. Within the floodplain (between the dikes), increasingly costly and marginally less efficient measures can contain drivers that make forward-looking multi-purpose land use difficult to achieve; this encourages efforts to extend the floodplain and find different ways of increasing the space available for the river.

The shift towards flood-peak polder design was also motivated by the cost of the alternative scenario of raising the dikes, which was three times more expensive than a polder system with a similar protective capacity [29]; this is considered as a "cost minimization" economic decision algorithm on investment alternatives with no monetization or comparison of benefits from alternatives.

Figure 1 shows polders constructed or upgraded between 2004 and 2017, including Tiszaroff, Nagykunság and Hanyi-Tiszasüly reservoirs, the dike relocation at Fokorú-

puszta, and the complex land management simulation at the Cibakháza-Tiszafüred former floodplain area.

Flood-peak polders are equipped to deal with the rarest of extreme flood events up over a 100-year return period when the cross-sectional discharge capacity is no longer sufficient to hold the flood waters between the dikes, even after temporarily raising them with sandbags to compensate for level deficits. Opening the flood gates provides controlled inundation that allows the peak of the flood wave to be cut off [30]; this is more effective compared to passive solutions such as dike level reduction toward the polder or dike relocation, where the excess water storage capacity is partly filled by water from the less threatening segment of the flood wave. The passive solutions have a lower flood risk reduction effect, either in terms of the additional area or the water quantity [31–33]; however, in terms of the potential for providing ecological services, controlled-operation flood-peak polders do not always fit with nature-based solutions. The extreme floods that would trigger the inundation of these polders, as originally designed, are too rare to provide the necessary water supply to the ecosystem. From an infrastructure management point of view, it is preferable to keep the arable farming that is located in the area and compensate for damage on a case-by-case basis [34]; this logic was implemented in the legislation that provides a framework for the use of the flood-peak polders of the Tisza River (2004/67).

The context, however, changes when the flood-peak polder opening is based on the economic balance of flood risk reduction gains over farmers' compensation for the damages in the polder inundated rather than the exhaustion of the cross-sectional capacity of the river corridor as a hydrological or defensibility trigger. The technical conditions for examining this question were not yet available when the polder system was designed, but now a more sophisticated cost-benefit approach (CBA) using quantified flood risk change methodology could be applied. The combined analysis of the individual outcomes can be used to compare the magnitude of public benefits to individual benefits in the case of the different land use and flood mitigation solutions applicable in the floodplains and the polders.

### 3.2. Single Purpose Flood Risk Reduction Performance of the Middle-Tisza Flood-Peak Polders

The first block of results summarizes the economic decision sphere of flood-peak polders' use and reveals the limitations of applying only one public benefit as an optimization criterion for land-use optimization.

Ref. [17] presented a detailed methodology for quantifying the flood risk mitigation effect of flood-peak polders with controlled inundation. The results prove that, from an overall societal perspective, it makes sense to allow more frequent use of flood peak polders, despite the significant event-based costs associated with inundation.

Using the polder to cut the peak of a flood provides the benefits of reducing flood risk and costs of flood defense activity downstream at the cost of compensation paid for agricultural damage caused by polder inundation. The relationship between the return frequency of the flood waves and the economic balance of the three polders in the Middle-Tisza section is illustrated in Figure 2. The results are based on a 50-year period using a discount rate of 2%. The economic break-even point for opening polders is located where the curves intersect the horizontal axis, representing the return period of a flood in which costs and benefits balance each other; these fall within the 20–25 year flood range return period compared to 100 years assumed on hydrological grounds. The concept of the economic break-even point creates a connection between the peak level flood in the river and the productivity of land use inside the flood-peak polder.

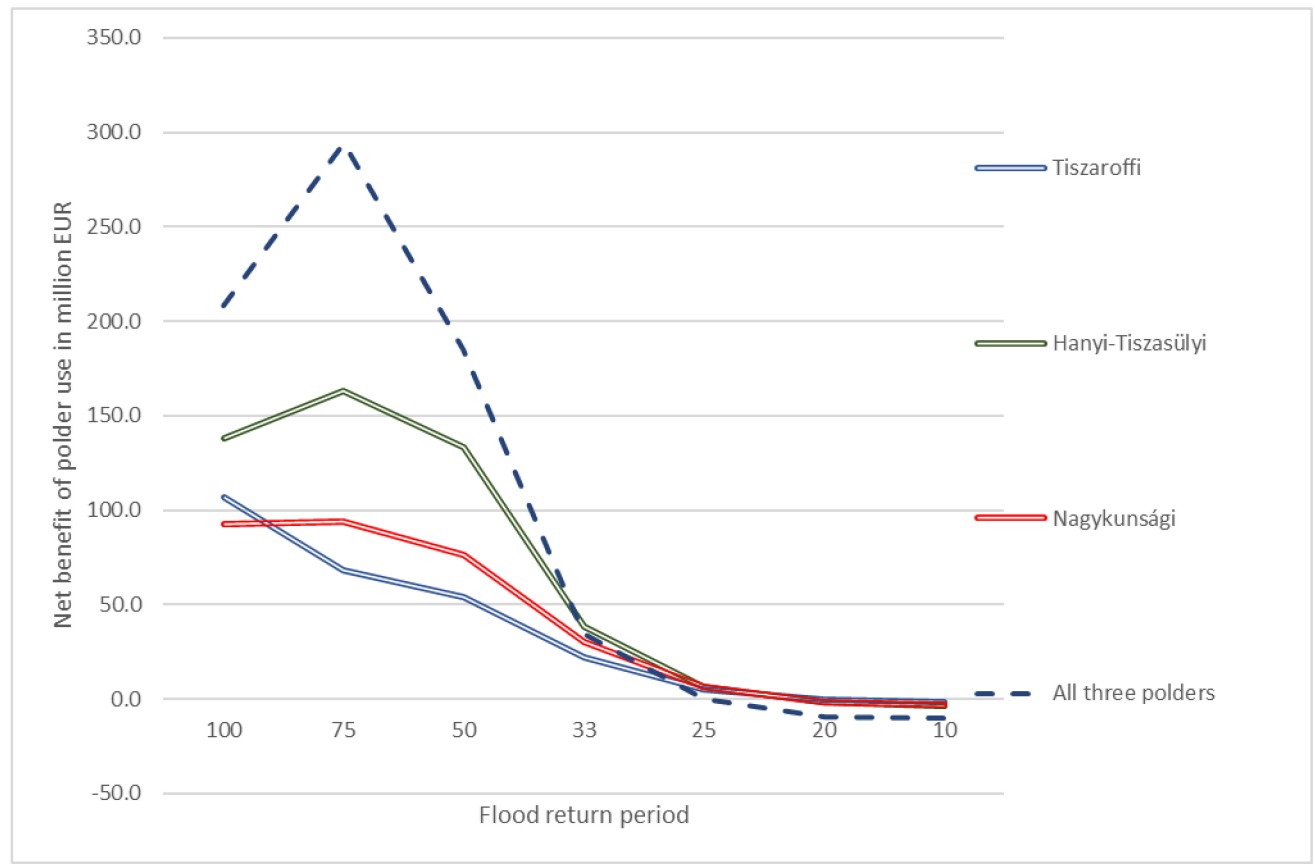

**Figure 2.** Middle Tisza flood peak polders' net benefit over different return period flood events. Source: Raw data of the figure with permission from [20].

In the longer-term, the economic breakeven point is affected both by the change in the value of the threatened properties downstream (typically an increase) and by the change in compensation for the inundation damage inside the polder. The latter can result from increased agricultural production intensity, which pushes up the economic breakeven point and leads to less frequent openings, or from changes that reduce exposure, such as more extensive land use, which has a downward effect on the economic breakeven point, leading to more frequent inundation. The productivity of land-use inside the polders therefore has an impact on the potential for risk reduction through opening the flood-peak polder.

Benefits from flood risk reduction in relation to land productivity can be quantified by aggregating the risk reduction effect that can be achieved with floods above the return period of the breakeven point (the area between a curve and the horizontal axis in 0). Table 1 shows the value of flood risk reduction as a 'service' per hectare of polder area. The first three columns describe the service values of the operation, ranging from EUR 18,000–34,000 per hectare.

**Table 1.** Flood peak polder area economic performance. Legend: EUR values calculated by the year 2020 average HUF/EUR exchange rate. Source of data [20].

| Polders | Value of Flood Risk Reduction Service in A 50 Year Time Period | Size of the Polder | Value of Flood Risk Reduction Service in A 50 Year Time Period/Hectare | Value of Flood Risk Reduction Service Below the Break-Even Point | Value of Flood Risk Reduction Service Below the Break-Even Point/Hectare | Price Of Land in The Region | Per Hectar Value of Flood Reduction Service Below the Break-Even Point/Price of Land |
|---|---|---|---|---|---|---|---|
| | Million EUR | Hectare | EUR/ha | Million EUR | EUR/ha | EUR/ha | Ratio |
| Nagykunsági | 72.6 | 4006 | 18,128 | 13.6 | 3400 | 4277 | 0.80 |
| Hanyi-Tiszasülyi | 99.4 | 5437 | 18,279 | 19.3 | 3545 | 4277 | 0.83 |
| Tiszaroffi | 81.3 | 2336 | 34,818 | 8.3 | 3548 | 4277 | 0.83 |

Under the assumption of lower damage exposure in the polder, the economic breakeven point also shifts downwards, providing an additional risk reduction effect for opening at lower flood levels; however, as flood waves decline the additional risk reduction effect also contracts. In a special case with zero compensation all floods can be released into the polder. In this case, however, current agricultural activities would no longer be viable in the area. The question arises as to whether the additional risk mitigation that could be gained by releasing minor floods would cover the cost of purchasing the land? (Land price is considered as a clear indicator of the economic space for bargaining with a landowner for mutually acceptable land-use terms and no policy considerations of expropriation are attached.) The fourth and fifth columns represent only the value of this additional benefit from minor floods; its relation to the land price in the area, the sixth column, reveals the economic viability of such an approach. The last column is sensitive to the length of the lifecycle and the discount rates that are close to but lower than one. In other words, the additional risk reduction benefits from floods below the breakeven point is lower than the costs of investing in the possibility of curbing them.

From a narrow flood protection point of view, therefore, the solution currently applied is confirmed by these ex-post calculations. Taking only flood risk into consideration, it is not economically rational in the current situation to change land-use, i.e., to invest in the purchase of land in the polder to further reduce the cost of flood events. At the same time, in order to reveal the full potential for additional public and private benefits, the bundle of benefits is necessary to be considered that the following two analyses pursue.

### 3.3. Multi-Purpose Cost-Benefit Relationships of Dike Relocation at the Fokorú-Puszta River Section

Ref. [21] carried out an ex-post evaluation of the dike relocation development at the Fokorú-puszta site. The flood risk reduction effect of the dike relocation overlaps strongly with the area of reduced flood risk by the flood-peak polders. While not substitutable, the flood risk reduction performance is comparable. Ref. [21] was based on the flood risk calculation methodology for polder use and complemented by an assessment of ecosystem-service-based benefits enabled by integrating the area into the floodplain.

The floodplain was extended by 325 hectares, converting what was previously arable cultivation into grassland, wetlands, and a forested strip to protect the dike (see Figure 3). The extended cost-benefit analysis of the investment and the multi-purpose use of the area quantified the flood risk reduction, the maintenance costs of the area, the revenue from the timber extraction and carbon sequestration balance from the forests, and the use of the wetland as a fish spawning area; these benefits from the full ecosystem service package are monetized and presented in Table 2.

**Table 2.** CBA results for ecosystem service elements. Legend: Source of data [21].

| | Size of the Delineated ES Service Area | Net Present Value of the ES Service |
|---|---|---|
| **CBA elements** | Hectare | Million EUR |
| Investment costs | | −15.8 |
| Flood risk reduction as benefit | 325 | 18.0 |
| $CO_2$ sink (forest) | 20 | 0.8 |
| Fish spawning area | 35 | 0.9 |
| Meadow maintenance | 270 | −0.5 |
| SUM | | 3.4 |

Flood risk reduction has by far the largest impact, itself bringing the equation into positive territory. The ecosystem services provided by the floodplain under current conditions (excluding flood risk reduction) would not cover the cost of the dike relocation project, but have a strong, positive supplemental impact.

The benefit of dike relocation and the polder development can also be expressed per hectare of land used. Table 3 shows the specific flood risk reduction impact per hectare of the three polders and the dike relocation.

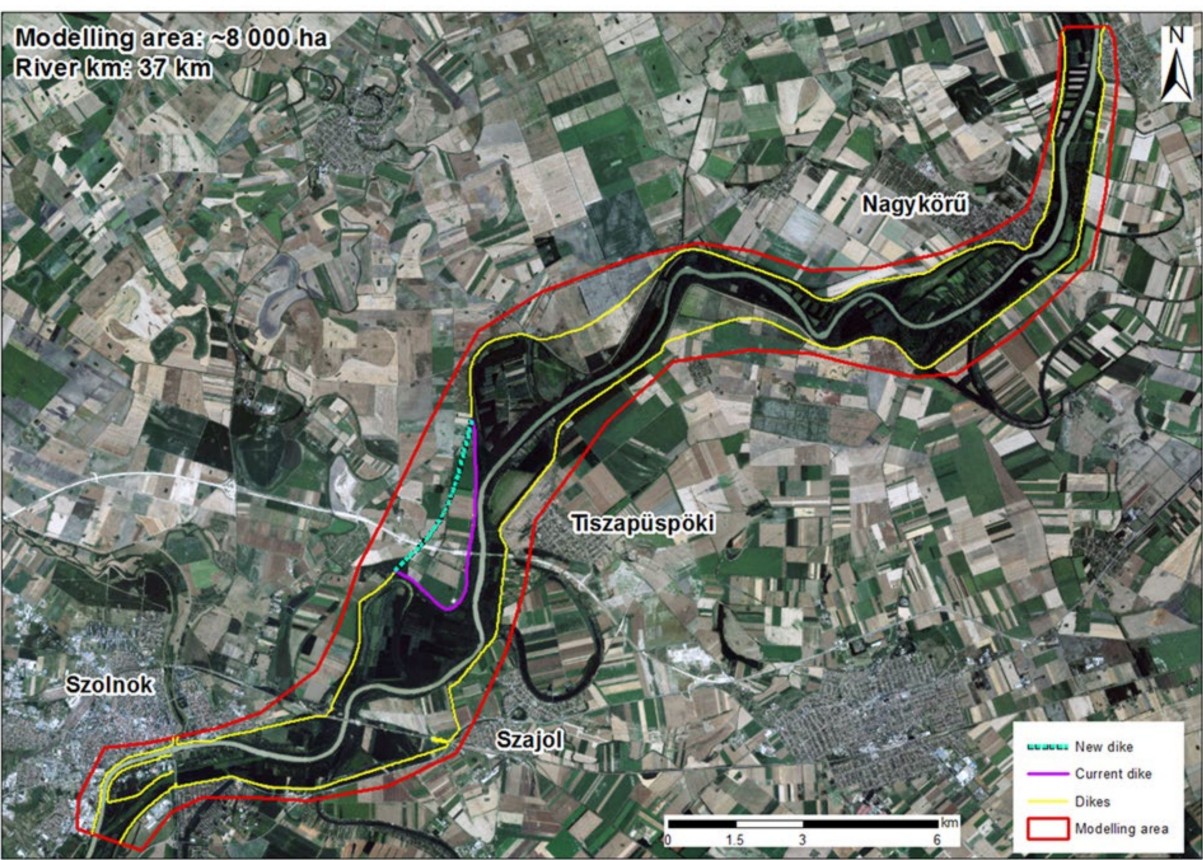

**Figure 3.** Map of the Fokorú-puszta dike relocation site. Legend: The map shows the change in the line of the dike and the area incorporated into the floodplain. Source: Ref. [21].

**Table 3.** Flood-peak polders and dike relocation land service productivity comparison. Legend: Source of data: Refs. [20,21].

| | | Impact of the Fokorú-Puszta Dike Relocation | |
|---|---|---|---|
| | | Flood risk mitigation impact<br>EUR/ha<br>6857 | Flood risk mitigation and other ESS impacts<br>EUR/ha<br>10,475 |
| **Flood risk mitigation impact of the flood-peak polders** | EUR/ha | Ratio of per hectare efficiency: dike relocation/flood-peak polder Only flood risk mitigation | Ratio of per hectare efficiency: dike relocation/flood-peak polder Including ES service of dike relocation |
| Nagykunsági flood-peak polder flood risk mitigation impact | 18,128 | 0.38 | 0.58 |
| Hanyi-Tiszasülyi flood-peak polder flood risk mitigation impact | 18,279 | 0.38 | 0.57 |
| Tiszaroffi flood-peak polder flood risk mitigation impact | 34,818 | 0.20 | 0.30 |

Table 3 shows that the dike relocation risk reduction capacity per unit area is much lower than that of the polders, a consequence of the controlled release of water. By opening flood gates, the top of the flood wave—which carries the most risk—is discharged into the area of the polder. In the case of dike relocation, the flood reduction effect comes from a local increase in cross-sectional runoff capacity, which becomes partly occupied by the volume of water ahead the peak of the flood wave. Therefore, the newly integrated area is not fully utilized to treat the most dangerous section of the flood wave. The last column shows how the differences contract when ecosystem services are included in the impact

of the dike relocation. Although the inclusion of additional ecosystem services does not overcome the initial difference, the results confirm the conclusion that the potential for developing multi-purpose use packages rather than considering single-purpose use.

Unlike the polder solution, relocating the dike includes the expropriation costs of the farmland concerned. The price of the land (present value of future income from its agricultural use) is covered by the total additional benefits raised by the project, mostly concentrated in flood risk mitigation; this can open the way for other uses based on the functioning of the ecosystem for smaller co-benefits that require the replacement of intensive farming.

It is important to highlight the role of forests, which have a significant added value relative to their share of the territory (6%); these benefits are calculated based on a management concept of continuous forest cover in line with the function of dike protection against waves; it is supplemented by the value of the additional carbon sequestration over the 50-year lifespan of the analysis; this follows the recommendation of Ref. [35] for a "shadow price of EUR 36–72 EUR/t$CO_2$" to increase atmospheric carbon dioxide concentrations (see details in the Appendix A).

The significant public benefits accrued from forests deserve additional attention in comparison with grasslands and wetlands. A significant obstacle to grassland benefits is the absence of pasture-based economic activities in the area of study where higher value-added benefits of raising and feeding animals could be materialized [21]. Maintaining grassland has delivered a near breakeven result with agricultural subsidies, mostly contributing through the reduction of flood risk. For wetlands, the negative balance is owing to the high cost of construction, which is a unique, location specific feature. The value of wetlands is predominately recreational fishing, but the value of the spawning areas would not increase proportionally since anglers would not proportionately increase the time they spend along the river. Under the present circumstances, forestry is the only land use type that would increase benefits in proportion to area expansion.

### 3.4. Combined Impacts of Flood Mitigation and Land Use Transformation in the Cibakháza-Tiszaföldvár Area

This third analysis [22] did not examine the operation of an already implemented infrastructure development, such as the other two. An additional conceptual study was executed, only this time without anticipating real life implementation; this multi-scenario analysis was carried out for the former floodplain area between Cibakháza and Tiszaföldvár settlements and now the area is protected from flooding. In addition to the current (flood-protected) agricultural land use scenario, two other land use scenarios with different exposures to damage were combined with three flood attenuation scenarios; this later analysis highlights the role of economic incentives driving land use decisions.

Farmland has a special role in managing flood risk, on the one hand benefiting from protection of the infrastructure in place, and on the other constraining the space for flood protection solutions in areas capable of adapting to transient water cover.

The choice of agricultural cultivation is an individual parcel-level decision, whereas flood risk mitigation can only be achieved through coordinated land use of a large area. Conversion would be rational, but implementation is hampered by a number of difficulties [36]. The crucial question for landowners is whether future income will rise or fall [37]. In the case of lower-income, individual compensation must be covered by the wider community benefits. For higher income, only implementation support for the collective conversion process should be covered by the community benefits. In the latter case, individual income distribution imbalances can be smoothed with assigned transaction costs over time; these hidden costs are significant, spread over many actors, and capable of undermining agreements [38].

In Hungary, the EU agricultural subsidies have an oversized influence on landowner decisions [39]. Even in the case of medium and poor-quality crop land, the subsidies provided under the Common Agricultural Policy dictate land use, making it economically

irrational to convert land for other agricultural purposes. Agricultural regulation is in a constant state of flux in order to avoid the direct and indirect negative environmental impacts it generates. Therefore, a recurrent analysis on the prospects of agricultural land-use, such as the area between Cibakháza and Tiszaföldvár, is necessary.

In the absence of dikes, the area would be regularly flooded by medium flood levels. The aim of the study was to investigate the financial opportunities and constraints for adaptation of agricultural land to flooding, considering both the agricultural damage caused by flooding and the flood risk reduction impact of different flood mitigation solutions. Can land-use adaptation create a positive balance taking into account all costs and benefits? Can benefits be allocated to landowners bearing the costs? The site selection criterion was not to maximize flood risk reduction but to explore the relationship between the land characteristics and its inundation.

Table 4 depicts the current land-use (Current LU) that is almost entirely arable (94%), with the first scenario (Adapted LU) crop-dominated (59%), adjusted to terrain and acceptable according to current public perception, and the second scenario fully forested (Forest LU).

**Table 4.** Land use ratios, combined scenario definitions and names.

| Land Use (LU) | Current LU | Adapted LU | Forest LU |
|---|---|---|---|
| Arable land (crops) | 94% | 59% | 0% |
| Grassland | 5% | 28% | 0% |
| Deciduous forest | 1% | 13% | 100% |
| **Total** (hectare) | 2067 | 2067 | 2067 |
| **Flood scenarios on LU scenarios** | | | |
| No flood, the area is protected | Current LU, no flood | Adapted LU, no flood | Forest LU, no flood |
| All floods, the area is open to inundations | Current LU, all floods | Adapted LU, all floods | Forest LU, all floods |
| Floods through a sluiceway above the 30 year return period floods | Current LU, spillway | Adapted LU, spillway | Forest LU, spillway |
| Polder-like operation, only the 100 year return period floods | Current LU, flood gate | Adapted LU, flood gate | Forest LU, flood gate |

The modelling scenarios attempt to cover the characteristic elements of the outcome spectrum. The three land-use options were tested in combination with the four flood scenarios. The hypothetical shift in land-use and introduction of inundation patterns is enabled by the construction of a perimeter dike, enclosing the area and connecting to the existing line of the dikes.

In the first scenario all floods reach the area ("all floods"), in the second scenario floods above the 30-year frequency are released through a lowered dike section ("spillway"), and in the third scenario the peak of the rarest floods (100-year return frequency) is cut by the flood gate ("flood gate"). The flood level reduction increases from the first to last scenario, with the latter similar to the operation of the controlled flood-peak polders previously described. The calculation accounts for Middle-Tisza flood-peak polders and the dike relocation since they are already influencing downstream flooding patterns.

Table 5 shows that under all scenarios frequent flooding damage to arable land cannot be compensated by the combined benefits of reduced flood risk, lower exposure, and increased ecosystem service potential from converted land. In other words, the investment costs are not offset by flood risk reduction and land-use change is insufficient in scale. Comparing values from the first and fifth columns, it shows, however, that the adapted land-use scenario is the more favorable without inundation.

**Table 5.** Costs and benefits of various scenarios in the Cibakháza-Tiszaföldvár floodplain area. Legend: The "Flood risk benefit minus agricultural damage" row of the table shows relative results, the BAU status has an initial flood risk exposure (−85 million EUR) that is excluded from the "Current LU, no flood" scenario for the better comparability. Source of data: Ref. [22].

| | Land Use (LU) and Flood Mitigation Scenarios | | | | | | | |
|---|---|---|---|---|---|---|---|---|
| | **BAU: Current LU—No Flood** | **Current LU, All Floods** | **Current LU, Spillway** | **Current LU, Flood Gate** | **Adapted LU, No Flood** | **Adapted LU, All Floods** | **Adapted LU, Sluiceway** | **Adapted LU, Flood Gate** |
| | Million EUR | Million EUR | Million EUR | Million EUR | Million EUR | Million EUR | Million EUR | Million EUR |
| Investment cost | 0.0 | −20.1 | −20.1 | −31.8 | 0.0 | −20.1 | −20.1 | −31.8 |
| Flood risk benefit minus agricultural damage | 0.0 | −18.2 | 0.8 | 8.0 | 0.0 | −10.8 | 1.2 | 8.5 |
| Agricultural activities' income | 8.7 | 8.7 | 8.7 | 8.7 | 8.5 | 8.5 | 8.5 | 8.5 |
| Financial transfers for agricultural activities | 14.7 | 14.7 | 14.7 | 14.7 | 14.1 | 14.1 | 14.1 | 14.1 |
| Value of $CO_2$ emission-sink balance | −4.5 | −4.5 | −4.5 | −4.5 | −1.5 | −1.5 | −1.5 | −1.5 |
| Sum | 18.8 | −19.4 | −0.4 | −4.9 | 21.2 | −9.7 | 2.3 | −2.1 |

One clear message from the modelling outcomes is that for crop based agricultural activities transfers substitute rather than complement individual income generation. Although EU-CAP subsidies (transfers) drive individual decisions on cultivation, long-run forestry is more beneficial than crop production, with or without transfers, as shown below in Table 6. In addition to subsidies, lower short-term annual income, lack of management experience beyond crop production, and the cost of replacing assets act as constraints that reduce the attractiveness of an otherwise superior long term financial outcome.

**Table 6.** Forest land use scenarios compared to business-as-usual (BAU) (Monetary values are net present values in million EUR. "Ratio x BAU" columns compare scenario results to the "Current LU, no flood" scenario. "Individual balance" refers to the land users' financial position with and without EU CAP payments. "Public benefits" are the sum of the flood risk reduction benefits and the shadow price of $CO_2$ sink. "Sum of all CBA items" includes all individual and public cost and benefits items together with the investment costs.) Source: Own calculations.

| | Land Use (LU) and Flood Scenarios | | | | | | | | |
|---|---|---|---|---|---|---|---|---|---|
| | **Current LU, No Flood (BAU)** | **Forest LU, No Flood** | **Forest LU, All Floods** | **Forest LU, Spillway** | **Forest LU, Flood Gate** | **Forest LU, No Flood** | **Forest LU, All Floods** | **Forest LU, Spillway** | **Forest LU, Flood Gate** |
| NPV results of selected items | Million EUR | Million EUR | Million EUR | Million EUR | Million EUR | Ratio x BAU | Ratio x BAU | Ratio x BAU | Ratio x BAU |
| Individual balance with Transfers | 23.4 | 32.7 | 32.7 | 32.7 | 32.7 | 1.4 | 1.4 | 1.4 | 1.4 |
| Individual balance without Transfers | 8.7 | 22.5 | 22.5 | 22.5 | 22.5 | 2.6 | 2.6 | 2.6 | 2.6 |
| Public benefits | −4.5 | 11.2 | 12.3 | 13.1 | 20.3 | −2.5 | −2.7 | −2.9 | −4.5 |
| Sum of all CBA items | 18.8 | 44.0 | 24.9 | 25.8 | 21.3 | 2.3 | 1.3 | 1.4 | 1.1 |
| Sum without $CO_2$ benefits | 23.4 | 32.7 | 13.7 | 14.5 | 10.0 | 1.4 | 0.6 | 0.6 | 0.4 |

However, the costs of preserving low value-added agricultural production go beyond the individual to the public (as a form of opportunity cost), which is made clear comparing Current LU scenario to the afforestation scenario (Forest LU).

Over several decades, forestry would yield 40% more individual income discounted into present value today compared to the current crop-dominated farming system, and it would be 2.6 times higher without agricultural subsidies (transfers). It is clear from Table 6 that the subsidies are the driving force for land-use. In order to unlock the superior public benefits from forestry rather than continue to focus on crop production, the income stream of farmers should be smoothed. Currently the subsidy system disincentivizes farmers from switching to forest management with attractive long-term prospects. In spite of the problem revealed the results mean that not continuous income support is necessary to induce land use adaptation, but a "bridging" type of public support is needed to focus on the organization issues and initiation of individual afforestation activities covering all land users.

We can see from the modelling results in the last column of Table 6 that without $CO_2$ mitigation the additional flood risk reduction service is not economical; this highlights the importance of the dual evaluation including $CO_2$ sequestration potential, which is 2.33 $CO_2$ ton/ha per year averaged over 50 years, and this value is the difference between the emission from arable cultivation and $CO_2$ sequestration of a forest, based on Hungary specific coefficients. In Table 6, the value of $CO_2$ sequestration is calculated using the mean value of the shadow carbon price used by the EBRD (see Appendix A for more details). Alternatively, Table 7 shows the breakeven $CO_2$ price for combined land-use change and flood protection (The EBRD methodology calls this the switching price of carbon).

**Table 7.** Carbon sink breakeven price in the forest scenarios. Source: Own calculations.

| | Forest LU, No Flood | Forest LU, All Floods | Forest LU, Spillway | Forest LU, Flood Gate |
|---|---|---|---|---|
| Balance: The sum of all items without $CO_2$ benefits compared to BAU (EUR/ha/year) | 144.2 | −148.9 | −136.1 | −205.3 |
| $CO_2$ emission difference between forest and arable (ton/ha/year) | −2.33 | −2.33 | −2.33 | −2.33 |
| Monetary value of $CO_2$ mitigation necessary for breakeven economic position of the scenario (EUR/$CO_2$ ton) | −62 | 64 | 58 | 88 |

All but the first scenario show a positive balance only with high $CO_2$ prices assumed since the flood risk reduction investments are not profitable without a price for $CO_2$; this EUR 64–88 per ton $CO_2$ price range, however, is near to current EU ETS prices [40] and EBRD and World Bank estimates (EUR 37–74/ton $CO_2$) [35].

The price orders of magnitude suggest that the public benefit of forestry in terms of $CO_2$ sequestration (excluding the flood risk impacts) is enough for landowners to shift from crop to forest management; however, this theoretical calculation for the Cibakháza-Tiszaföldvár site shows that, if $CO_2$ avoidance prices are assumed, a positive balance of combined flood risk reduction and afforestation in the former floodplain areas of the Tisza River basin can be achieved by more risk-efficiency-focused location and size choices.

*3.5. Linking of the Research Results*

The combined analysis aims to reveal the complex territorial and sectoral interrelationships and their potential for joint optimization. The public benefits can be characterized as purely flood risk reduction as current peak polder land-use fails to realize their full potential. Albeit from this risk-only perspective, it is not worth transforming the current polder land-use at the cost of buying the land—the individual and public benefits of afforestation even separately would cover this difference as the calculations of the Cibkaháza-Tiszafüred site show.

The calculations for the Fokorú-puszta dike relocation project indicate that the balance of the floodplain expansion would be significantly improved if new ecosystem service opportunities could be developed to complement the flood risk reduction effect.

The results of the study for the Cibakháza-Tiszaföldvár area makes a compelling case for forestry as opposed to crop-dominated farming. With short-term agricultural incentives, it would be rational to maximize both individual and community benefits. While the forestry brings higher long-term individual income, it is dampened by the short-term incentives of the agricultural subsidy regime. A shift towards the provision of community benefits requires management of the land-use transition, not agricultural income substitution in perpetuity. State funding for farmers should be reassessed to overcome the challenges of the transition and develop the foundation for merging benefits over time.

The economic exploration of trade-offs between flood risk change and afforestation necessitates their joint optimization based on studying out-of-the-box hydrological scenarios without restrictions on river corridor roughness and width conditions and the frequency of

polder inundation; these novel hydrology simulations can reveal the flood risk limitations of transforming the space and time distribution of flood wave profiles.

## 4. Discussion

The quantified flood risk and forestry $CO_2$ sequestration assessments carried out in this study are underused for the implementation of public policy in Hungary. The main problem for flood-peak polders with controlled inundation is the low relative frequency of their use, maintaining arable cultivation [34], which comes with the opportunity cost of the potential for additional benefits. Forest areas can minimize the inundation cost and additional benefits can be obtained. An example of this approach is the decades long experience of the Mályvád polder at the Körös River [41], where both flood damage prevention and water management infrastructure elements have been technically installed. The area is regularly replenished from receding floods and peaks can be cut-off by flood gates. So far, only ecological studies have been carried out [42] and a more complex evaluation is needed. There is room for technical innovation from both an engineering and a land management perspective to ensure that water can be safely diverted from regular floods. There is also a need to explore the cost-benefit impacts of flooding on forested areas through empirical studies of local flood and seasonal water inundation conditions. Recently the national methodology for risk mapping under the EU Floods Directive protocol assumes damage functions for both regular and hydrologically affected forests [43].

The results show that there is a strong public interest in the accountability of the offsetting $CO_2$ sequestration capacity of floodplain forests. A transparent and therefore widely considered accounting system that improves incentives to conserve forests has significant potential for achieving public policy objectives across sectors [44]. At the same time, as in the context of flood risk management, there is a key role for private landowners to play in the development of nature-based solutions [45]. To realize the potential of $CO_2$ sequestration, it is necessary to clarify who owns the $CO_2$ sequestered as a climate mitigation service [46]. At present, $CO_2$ sequestration from forests contributes to the national obligations, e.g., in Hungary, involving emissions not covered by other regulations, where it is the responsibility of the government to achieve emission levels in line with international commitments. There is no feedback to forest owners to steer their management decisions in a direction that benefits the community or to give them the right to sell $CO_2$ sequestration. It is not clear whether forest owners can be given the right to sell their sequestration in a voluntary carbon offsetting scheme without creating a double-counting with national compliance. Uncertain property right arrangements hamper innovation. The increasingly apparent value of $CO_2$ sequestration and the scale of the potential areas involved for afforestation suggest a public benefit be harnessed justifies deeper legal focused investigation, which Ref. [47] outlines.

The economic interpretation points to a hitherto unstudied hydrological relationship between more frequent flood-peak polder utilization and higher roughness in the floodplain. Can this flood risk transfer work? The additional risk mitigation capacity of polders may improve management conflicts that arise between the most ecologically valuable riparian gallery forests and the declining cross-sectional capacity; this approach opens new hydrological analysis opportunities for higher frequency polder usage in flood risk regulation considering individual and public benefits of ecological land management.

In the long run, the adaptability of land-use in the former floodplain areas is essential in the context of a landscape that is gradually losing water [48]. The multidirectional, dispersed damage of droughts is already more costly than flooding [49,50], but detailed, site-related information is not yet available. The quantified benefits of land-use adaptation processes are also important as they facilitate infiltration by the more frequent inundations. With the right tree species, the water tolerance offers flexibility to shift the boundary of the area under protection; this in turn has repercussions on the spatial constraints of flood risk management itself and the ability to expand the cross-sectional capacity of the river, which has been minimized for the Tisza River to this point.

## 5. Conclusions

The modelling results show that the improvement of ecosystem service based social benefits of riparian areas must be based on an integrated approach for flood risk reduction and land adaptation rather than fragmented sectorial and location boundaries.

The benefits of the quantified flood risk management method and the benefits of accounting for $CO_2$ sequestration of forests, even conservatively calculated as two robust and transparent methodologies for quantifying impacts, provide sufficient information to inform about the economics of land-use change and adaptation processes; this creates the economic conditions for the necessary agreements between the community of beneficiaries and the cost bearers. With this approach, the issue of agreements on subsequent uses that may potentially arise when ecosystems mature can be left for later.

The community cost of not adapting riparian land-use far outweighs the opportunity costs of individual inaction. The price of land as a switching point for expropriation is a clear indicator in an economic comparison, moreover it indicates a strong public bargaining position on forming future land use, but the displacement of land-users is not the preferable negotiation result. Overcoming the underlying conflict of interests may require a legal mandate for the creation of a compulsory water easement regulation that enable the majority of agricultural landowners of a local flood basin to receive transient water cover that the recent requirement of a unanimous owner agreement would never be reached. Due to the perverse incentives imbedded in the EU-CAP subsidies, there is a need for sufficiently strong and enforceable regulatory instruments to enable community action on long-term water management goals.

In order to increase the social benefits from forested $CO_2$ sequestration capacity, the principles of public and private ownership and a national accounting system need to be clarified and established.

**Funding:** This research received no external funding.

**Institutional Review Board Statement:** Not applicable.

**Informed Consent Statement:** Not applicable.

**Data Availability Statement:** The data that support the findings of this study is available from the corresponding author upon reasonable request.

**Acknowledgments:** I thank my colleague András Kis for his enduring support.

**Conflicts of Interest:** The author declares no conflict of interest.

## Appendix A

The World Bank and many large international development banks already use the projection of the "carbon price" in their financing decisions for project approval [35]. It is assumed that carbon pricing will become standard practice within the timeframe of the projects analyzed. Until then, there are several ways to estimate the economic value of carbon dioxide sequestration, but they vary widely. The most transparent global $CO_2$ price is the EU-Emission Trading System (ETS), which is EUR 78/ton at the moment, but has fluctuated between EUR 16/ton and EUR 96/ton over the last two years (EUA Futures, 2022). Other non-equivalent carbon markets provide less reliable price information. Another approach is equating the economic value of $CO_2$ emissions (or removals) to a price range that will phase out fossil-based technologies in order to meet the Paris Climate Goals; this approach is used by the World Bank and a number of major international investment banks to assess project economics. The price range is USD 40–80/ton $CO_2$ now, (i.e., EUR 36–73/ton) rising to USD 50–100 per ton by 2030, and then 2.25% per year until 2050.

According to the EBRD methodology, when evaluating projects, the lower and upper values of the price range are used to assess the project's outcome. If the balance of the proposed project is positive even after taking into account the cost of $CO_2$ neutralization at these prices, it is eligible for funding according to its climate protection performance.

If the calculation does not show a positive balance with both values, a switching point is calculated, whose value and the specific $CO_2$ abatement costs for the economy in question provide the basis for the financing decision.

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
