# Peer review of "Combining Flood Risk Mitigation and Carbon Sequestration to Optimize Sustainable Land Management Schemes: Experiences from the Middle-Section of Hungary’s Tisza River"

_land, doi:10.3390/land11070985_

Round 1

Reviewer 1 Report

Dear authors,

Thanks for the very interesting manuscript, which is not that common, regarding the topic covered. It was very learningful, and I found it of scientific interest especially for water resource managers at the basin/watershed scale.

I have no comment to add. I recommend accepting the paper in its present form.

Author Response

Dear Reviewer,

Thank you for your affirmative opinion.

Sincerely,

Gábor Ungvári

Reviewer 2 Report

I think the word "title" should be deleted from the title.

In the abstract, it would be advisable to include information about the results obtained or what this study aims to do.

In the article some errors appear in paper or figures citation.

Table 2017? You may be referring to Figure 1.

The introduction only gives us information about the events on the Tisza. But I would have expected an article to contain in the introduction information about the global and regional situation of the studied subject. Only then to discuss the local situation. I recommend that the introduction be revised.

I believe that the article needs to be reorganized. I would also like to include the classic division of type, materials and methods, results. I don't understand what some subchapters are about. Such as the chapter: "The transformation of flood defense strategy along the Tisza River", this is part of the introduction, study area, materials and methods.

Figure 2 is taken from: https://rekk.hu/downloads/projects/DanubeFloodplain_Hungary_Tisza_CaseStudy_REKK_03.04.2020.pdf

I assume that an article should contain original images or at least quote after the image from where they are taken.

"Article (XX, 2022)" - this is a citation?

What modeling scenarios were used: Combined impacts of flood mitigation and land use transformation in the Cibakháza-Tiszaföldvár area? I would have liked to know how the data in Table 5 were obtained. What are the calculations on which the statements in the article are based?

The title is confusing, I expected it to be an article based mainly on the analysis of carbon sequestration in the soil. But more than half of the article talked about the benefits of making dams and polders and whether it is economically beneficial to use the land in the floodplain for agricultural purposes.

Author Response

  1. I think the word "title" should be deleted from the title.

- Done

  1. In the abstract, it would be advisable to include information about the results obtained or what this study aims to do.

– Abstract is rephrased. Aims at row 15-18. Main takeaways row 19-30

  1. In the article some errors appear in paper or figures citation.

 – These were the effect of the template transformation of the submission process. They are addressed.

  1. Table 2017? You may be referring to Figure 1.

– Same as point 3.

  1. The introduction only gives us information about the events on the Tisza. But I would have expected an article to contain in the introduction information about the global and regional situation of the studied subject. Only then to discuss the local situation. I recommend that the introduction be revised.

 – A starting paragraph is inserted into the Introduction, row 34-56

  1. I believe that the article needs to be reorganized. I would also like to include the classic division of type, materials and methods, results. I don't understand what some subchapters are about. Such as the chapter: "The transformation of flood defense strategy along the Tisza River", this is part of the introduction, study area, materials and methods.

- A Data and Methods chapter is included (row 93-140)

  1. Figure 2 is taken from: https://rekk.hu/downloads/projects/DanubeFloodplain_Hungary_Tisza_CaseStudy_REKK_03.04.2020.pdf , I assume that an article should contain original images or at least quote after the image from where they are taken.

– Figure is referenced. Row 278. The sources used are addressed in the Data and Methodology section and in the Cover letter for the Editor

  1. "Article (XX, 2022)" - this is a citation?

 – It was described in the Cover letter for the Editor that there is one article at the very end of its publication process. I expected to be solved during the review process.

  1. What modeling scenarios were used: Combined impacts of flood mitigation and land use transformation in the Cibakháza-Tiszaföldvár area? I would have liked to know how the data in Table 5 were obtained. What are the calculations on which the statements in the article are based?

- The scenario combinations are detailed in Table 4. In the new chapter Data and Methods (row 93-140) the logic of the article is described. Because the article’s aim is to draw conclusions from the combination of the different results it doesn’t go into details of their sub-calculations, that is beyond the scope of this paper, meanwhile all these elements are openly accessible.

  1. The title is confusing, I expected it to be an article based mainly on the analysis of carbon sequestration in the soil. But more than half of the article talked about the benefits of making dams and polders and whether it is economically beneficial to use the land in the floodplain for agricultural purposes.

– I’m sorry for that it might occur due the expression as „land management schemes” has a much diverse understanding than the narrow ones each of us have in mind. Would a change in the title to “land use schemes” help to avoid this misunderstanding?

Reviewer 3 Report

Please format the article following MDPI templates and add row numbers before resubmitting.  

Author Response

The submission interface allows uploading material for publication in word format. Unfortunately, after uploading, the format conversion generated a number of errors at a later stage of the process. These have been corrected. As the journal format was not a pre-requisite for submission and would have taken a significant amount of time to reformat at this stage, it is reasonable that a full format correction should be made after a (hopefully) positive decision. I believe that the text in its current format is suitable for content assessment.

I also re-submit a pdf version to avoid further misunderstandings.

Reviewer 4 Report

The introduction provides a background to the topic, especially connected with the changing flood protection strategy in the affected area. The results are based on analyses that evaluate the areas at risk of floods from an economic point of view. The work is based on the results of research that has already been published and probably for this reason is not devoted to a separate chapter with the methodology used. The author's intention is clear, but it is unusual to dissolve the methods between the introduction and the results. In addition, the role of the author in the studies used, and at least quite marginally the main methods on which the work is based could be mentioned. The article lacks citation in at least two places (XX, 2022), which is problematic as the results refer to these publications. In Figure 3, there is a map containing orthophotos, which are not cited in the figure or in the text.

Round 2

Reviewer 2 Report

I consider that tables should be inserted as a table, not an image (printscreen from excel).

Otherwise I accept the changes made.

Author Response

Dear Reviewer,

Thank you for your comments. I re-inserted the tables a excel objects.

I have harmonized the language style to match the requirements of the US English and changes were made to improve the readability as well.

Kind regards,

GU

Reviewer 3 Report

This manuscript shows great improvement and can be considered as publishable.

Author Response

Dear Reviewer,

Thank you for your comments and your positive opinion on the improvements made. 

I harmonized the language according to the US-English requirements and applied some minor changes.

Kind regards

GU

Reviewer 4 Report

The added and modified parts of the text have addressed the most significant shortcomings of the article.

Author Response

Dear Reviewer,

Thank you for your positive opinion on the applied improvements. 

Kind regards,

GU